# Wood Degradation by *Fomitiporia mediterranea* M. Fischer: Exploring Fungal Adaptation Using Metabolomic Networking

**DOI:** 10.3390/jof9050536

**Published:** 2023-04-30

**Authors:** Marion Schilling, Marceau Levasseur, Muriel Barbier, Lydie Oliveira-Correia, Céline Henry, David Touboul, Sibylle Farine, Christophe Bertsch, Eric Gelhaye

**Affiliations:** 1INRAE, IAM, Université de Lorraine, 54000 Nancy, France; 2CNRS, Institut de Chimie des Substances Naturelles (ICSN), UPR2301, Université Paris-Saclay, Avenue de la Terrasse, 91198 Gif-sur-Yvette, France; 3INRAE, AgroParisTech, Micalis Institute, PAPPSO, Université Paris-Saclay, 78350 Jouy-en-Josas, France; 4CNRS, Laboratoire de Chimie Moléculaire (LCM), UMR 9168, École Polytechnique, Institut Polytechnique de Paris, Route de Saclay, 91128 Palaiseau, France; 5Laboratoire Vigne Biotechnologies et Environnement UPR-3991, Université de Haute-Alsace, 33 Rue de Herrlisheim, 68000 Colmar, France

**Keywords:** *Fomitiporia mediterranea*, *Trametes versicolor*, adaptation, white rot, molecular network, grapevine wood, Esca

## Abstract

*Fomitiporia mediterranea* M. Fischer (Fmed) is a white-rot wood-decaying fungus associated with one of the most important and challenging diseases in vineyards: Esca. To relieve microbial degradation, woody plants, including *Vitis vinifera*, use structural and chemical weapons. Lignin is the most recalcitrant of the wood cell wall structural compounds and contributes to wood durability. Extractives are constitutive or de novo synthesized specialized metabolites that are not covalently bound to wood cell walls and are often associated with antimicrobial properties. Fmed is able to mineralize lignin and detoxify toxic wood extractives, thanks to enzymes such as laccases and peroxidases. Grapevine wood’s chemical composition could be involved in Fmed’s adaptation to its substrate. This study aimed at deciphering if Fmed uses specific mechanisms to degrade grapevine wood structure and extractives. Three different wood species, grapevine, beech, and oak. were exposed to fungal degradation by two Fmed strains. The well-studied white-rot fungus *Trametes versicolor* (Tver) was used as a comparison model. A simultaneous degradation pattern was shown for Fmed in the three degraded wood species. Wood mass loss after 7 months for the two fungal species was the highest with low-density oak wood. For the latter wood species, radical differences in initial wood density were observed. No differences between grapevine or beech wood degradation rates were observed after degradation by Fmed or by Tver. Contrary to the Tver secretome, one manganese peroxidase isoform (MnP2l, jgi protein ID 145801) was the most abundant in the Fmed secretome on grapevine wood only. Non-targeted metabolomic analysis was conducted on wood and mycelium samples, using metabolomic networking and public databases (GNPS, MS-DIAL) for metabolite annotations. Chemical differences between non-degraded and degraded woods, and between mycelia grown on different wood species, are discussed. This study highlights Fmed physiological, proteomic and metabolomic traits during wood degradation and thus contributes to a better understanding of its wood degradation mechanisms.

## 1. Introduction

Eutypiose, Black Dead Arm and Esca are the three main grapevine trunk diseases worldwide, causing up to 50% of vineyards production and more than 1.6 billion dollars of economical loss [1]. One of the typical symptoms of Esca is white rot in the trunk, also called amadou [2]. In Europe, *Fomitiporia mediterranea* M. Fischer (*Basidiomycota*, *Hymenochaetaceae*) (Fmed) is the main fungal species identified in amadou [3,4,5]. Fmed is a wood-decaying fungus, classified as a white rot, as it is able to mineralize all wood structural compounds, including lignin, the most recalcitrant one [6]. Despite no true evidence of Fmed pathogenicity *stricto sensu* on grapevine, Fmed abundancy in grapevine wood was correlated with Esca foliar symptoms in vivo [7]. Understanding Fmed wood degradation mechanisms and adaptation to its ecological niche is thus crucial for a better knowledge of Esca.

Few studies also reported Fmed presence on other hardwood species [5], such as oak and beech wood [8]. However, its large presence in vineyards suggest that its molecular mechanisms are particularly adapted to degrade grapevine wood. Wood-decaying fungi developed substrate-specific degradation mechanisms, for example depending on wood, their substrate, chemical and anatomical traits. Thus, they might be well adapted to their substrate. Grapevine wood chemical and structural compositions show specificities compared to forest tree woods [9]. Fmed adaptation to such a specific wood composition, regarding its wood degradation mechanisms, remains poorly studied.

Wood decay involves wood structural compounds degradation, i.e., hemicelluloses, cellulose and lignin. Moreover, in response to microbial attack, grapevine wood is able to produce or to accumulate specialized metabolites (SMs) involved in the plant defense reaction against pathogens such as fungi, also called phytoalexins [10,11]. In wood, SMs are parts of the extractives, i.e., all molecules not physically bound to wood cell wall polymers and extractible with solvents [12]. These extractives belong to diverse chemical families such as stilbenes, terpenes, flavonoids or tannins [13], the most represented in grapevine wood being the stilbenes. To counteract with extractives, wood-decaying fungi, such as Fmed, are able to detoxify them, by degrading or modifying them chemically [14]. More specifically, Fmed was shown to degrade extractives in vitro in grapevine and beech wood sawdust, i.e., stilbenes and flavonoids [15]. Fmed degradation mechanisms for those extractives remain unknown but could involve extracellular laccases or peroxidases [16,17,18,19]. For example, one Fmed laccase has been shown to degrade resveratrol, the main grapevine stilbene monomer, and to oxidize, dimerize or modify stilbenes [16,20,21,22]. It is not known whether Fmed uses specific detoxification mechanisms depending on its woody substrate.

Other SMs can be produced by fungi and play crucial and diverse roles in survival functions, such as pathogenicity [23], stress response [24], micro-nutrients availability [25] or signaling. According to their involvement in environmental stress response, fungal SMs are often not produced by fungi under laboratory growth conditions. However, in vitro cultures with plant-derived media were shown to promote fungal SMs production [26]. Fmed was shown to produce SMs with phytotoxic activities in vitro, the 4-hydroxybenzaldehyde and the 6-formyl-2,2-dimethyl-4-chromanone, isolated during Fmed growth on carrot broth [27]. Their role in Fmed growth and pathogenicity in vivo is unknown, and such fungal SMs in wood colonized by Fmed were never reported in vineyards. 

Non-targeted metabolomic analysis using molecular networking was successfully used to investigate both wood and fungal SMs [28,29]. Molecular networking was first introduced in 2012 [30] and is a tool using tandem mass spectrometry (MS/MS) to visualize the diversity of a complex mixture of metabolites. Similar methodologies were already used mainly for protein annotation and pathways identifications, for example, with the open-source software Cytoscape [31]. The principle relies on the fact that two compounds that have similarities in their structure will have common fragment ions and/or neutral loss between their parent and fragment ions [32]. MS/MS spectra are compared using their cosine score and a clustering algorithm [33]. The cosine score indicates the degree of similarity between two MS/MS spectra, considering their parent and fragment ions [32]. Global Natural Products Social Molecular Networking (GNPS, https://gnps.ucsd.edu/, accessed on 26 April 2023) is a community providing tools and sharing databases for molecular network analyzes and annotations [34]. The open-source software MetGem (https://metgem.github.io, accessed on 9 March 2022) is an interface for applying this method and using the available databases [35]. MetGem proposes not only the molecular network visualization but also the *t*-distributed Stochastic Neighbor Embedding (*t*-SNE) algorithm, which preserves distances between clusters, among other advantages [35]. This method was successfully used for the characterization of new metabolites from microorganisms with the help of complementary techniques such as nuclear magnetic resonance [28,29].

The aim of this study is to describe Fmed metabolism depending on the presence of different wood species, including grapevine wood. Particular attention will be given to the evolution of wood structural and chemical composition (i.e., wood cell wall polymers and extractives content) during fungal colonization. To eventually highlight Fmed specific wood degradation mechanisms, sterilized wood blocks of three different wood species (grapevine, oak and beech) will be used. Wood degradation rates and fungal biomass production on different woody substrates will be compared, and wood fibers analyses, proteomics and non-targeted metabolomics with molecular networking will be used to investigate molecular mechanisms. The well-studied white-rot fungus *Trametes versicolor* (L.) Lloyd (*Basidiomycota, Polyporaceae*) (Tver) will be used as a comparison model to highlight fungal-substrate specificities. 

## 2. Materials and Methods

### 2.1. Fungal Cultures

Fmed strains phco36 and LR124 were provided by the Laboratoire Vigne Biotechnologies et Environnement (LVBE) of the University of Haute-Alsace in Colmar and isolated, respectively, from *Vitis vinifera* L. cultivar (cv.) ‘Ugni blanc’ in 1996 in Saint-Preuil (Charente, France) and from *V. vinifera* L. cv. ‘Carignan’ in 1996 in Villeneuve-lès-Maguelone (Hérault, France). Tver strain BRFM1532 was isolated from *Fagus* sp. dead wood in France and obtained from the Centre International de Ressources Microbiennes (CIRM) catalogue. For each strain, four 0.5 cm^3^ inocula were inoculated on 50 mL Malt-agar medium (20 g/L) in 175 cm^2^ cell culture flasks at 25 °C. After 2 weeks, three wood blocks per flask were added on the mycelium. Five flasks for each wood species (beech, oak and grapevine) on each fungal strain (two Fmed strains, one Tver strain) were cultured at 27 °C for 7 months. Oxygen in these 45 flasks was renewed once a week by opening the flasks under sterile conditions for one hour. Three flasks for each wood species on each fungal strain, and one flask with each fungal strain without wood (for control mycelium), were cultured at 25 °C for 4 months. Oxygen in these 30 flasks was renewed constantly through filtering caps and humidity was maintained at 80%. Mycelia collected after 7 months were lyophilized and weighed for fungal biomass estimation. 

### 2.2. Wood Material

Beech (*Fagus sylvatica*) and oak (*Quercus robur*) wood blocks were provided by the Biogéochimie des Ecosystèmes Forestiers (BEF) laboratory (INRAE, Champenoux) from healthy sapwood of trunks from Champenoux forest (Meurthe-et-Moselle, France). Entire vines of *V. vinifera* cv. ‘Riesling’ were cut in Orschwihr vineyard in Haut-Rhin (Alsace, France) in February 2021, and stored at room temperature before being sawn into wood blocks. White parts of the vine trunk, considered as healthy, were selected as much as possible. For each wood species, wood blocks of approximately 1.25 cm × 5 cm × 1 cm were sawn, dried 48 h at 105 °C, individually identified, measured (length, width and height) and weighed. Initial wood density was obtained for each wood block by the following equation: (1)$$wood\;block\;density=\frac{wood\;block\;dry\;weight\;\left(g\right)}{wood\;block\;volume\;\left({\mathrm{cm}}^{3}\right)\times \;\mathrm{water}\;\mathrm{volumic}\;\mathrm{mass}}$$
where density is a dimensionless value and water volume mass is equal to 1.00 g/cm^3^.

Mycelia on the surface of the wood and degraded wood blocks containing mycelium (grown within the wood) were collected separately either after 4 months for proteomic, metabolomic analysis and wood fibers analysis, or after 7 months for wood mass loss, fungal biomass estimation and wood fibers analysis. To measure wood mass loss, wood blocks collected after 7 months were dried for 48 h at 105 °C and weighed.

### 2.3. Wood Fiber Analyses

Wood fibers analyses were performed by ASIA platform (Université de Lorraine-INRAE; https://a2f.univ-lorraine.fr/en/asia-2/, accessed on 26 April 2023). Dried wood blocks collected after 4 and 7 months of culture were grown into powder by liquid nitrogen-assisted cold ball grinding (Cryomill Retsch^®^, Eragny sur Oise, France) and pooled by strain and wood species. Lignin, cellulose, hemicelluloses and soluble compounds contents were quantified as described previously [15], using a protocol adapted from Van Soest and McQueen [36]. 

### 2.4. Proteomic Extraction and Analyses

Wood blocks from 4 months cultures were cut into 1 cm^3^ pieces, and 6.00 g fresh weight of pooled wood blocks from each flask were extracted with 40 mL of potassium acetate buffer 30 mM pH 4.5 at room temperature at 60 rpm for 2 h. Extracts were centrifuged at 4000 rpm for 20 min at 4 °C and 10 mL of supernatant was precipitated in 80% cold acetone one night at −20 °C. After centrifugation (4000 rpm for 20 min at 4 °C), acetone and buffer were removed and dried pellets were dissolved in 90 to 220 µL of concentrated Laemmli buffer (2-mercaptoethanol 24% (*w*/*v*), bromophenol blue 17.3 M, glycerol 20% (*v*/*v*), sodium dodecyl sulfate 10% (*w*/*v*) and Tris-base 0.4 M). Samples were heated 10 min at 90 °C and 15 to 25 µL were deposited on 15% 1 mm SDS-PAGE gels. Bulks for all samples and one for each fungal species were constituted to be deposited on gel as well. After a short migration and Coomassie blue coloration, gel pieces were sent to PAPPSO platform (http://pappso.inrae.fr, accessed on 26 April 2023) for trypsin digestion and UHPLC-MS/MS analyses. For functional analysis, proteins found in all samples were selected upon their abundance, and only proteins for which the exponentially modified protein abondance index (emPAI) [37] was higher than 1 were selected. Relative abundances of the latter were estimated using normalized spectral abundance factor (NSAF) [38].

### 2.5. Ethyl Acetate Extraction and LC-MS/MS Analysis

Sixty ethyl acetate extracts were obtained from 24 conditions: dried and ground wood blocks of each of the three wood species (grapevine, beech, oak), after 4 months of exposure to each of the three strains (Fmed phco36, Fmed LR124 or Tver BRFM1532), and each corresponding mycelium (18 conditions) was extracted in triplicates (54 extracts). Nine dried, ground and pooled non-degraded wood blocks of each wood species (control wood) and mycelium of each fungal strain grown 4 months on malt-agar without wood (control mycelium) were extracted and used as control (6 extracts). The samples (wood blocks and mycelium) were dried for 48 h at 35 °C before being ground by liquid nitrogen-assisted cold ball grinding (Cryomill © Retsch, Haan, Germany) for ethyl acetate extraction. The powder was extracted at a ratio of 100 mg/2 mL with ethyl acetate by 24 h of maceration at room temperature, under dark conditions and at 60 rpm. Samples were centrifugated at 13,400 rpm and the supernatant was collected, dried at room temperature and weighed. Methanol was added to resuspend dried extracts at 1 mg/mL. Samples in methanol were additionally centrifugated at 13,400 rpm and the supernatant was collected, dried at room temperature and weighed. Methanol was used to resuspend dried extracts at 1 mg/mL for HPLC-MS/MS analysis. HPLC-MS/MS analysis was conducted at the ICSN mass spectrometry laboratory using Agilent HPLC-QTOF-ESI-MS/MS in positive ionization mode. 

### 2.6. HPLC-QTOF-ESI-MS/MS Analysis

High-performance liquid chromatography (HPLC) was performed using a 1260 Prime HPLC (Agilent Technologies, Waldbronn, Germany) with an Accucore RP-MSC18 column (100 mm × 2.1 mm × 2.6 µm, Thermo Scientific, Les Ulis, France). The oven temperature was set at 45 °C. A volume of 5 µL per sample was injected and the mobile phase was composed of A: water acidified with formic acid at 0.1%; and B: acetonitrile with formic acid at 0.1%. The gradient was performed at 0.4 mL/min with 95% A to 0% A for 20 min then maintained for 3 min, followed by an equilibration step with 95% A for 1 min then maintained for 4 min. Mass spectra were recorded in positive ion mode, in data-dependent analyses (DDA), with the following parameters for electrospray ionization (ESI): gas temperature 325 °C, drying gas flow rate 10 L/min, nebulizer pressure 30 psi, sheath gas temperature 350 °C, sheath gas flow rate 10 L/min, capillary voltage 3500 V, nozzle voltage 500 V, fragmentor voltage 130 V, skimmer voltage 45 V, octopole 1 RF voltage 750 V. For ESI, internal calibration was achieved with two calibrants, purine and hexakis (*m*/*z* 121.0509 and *m*/*z* 922.0098, respectively), providing a high mass accuracy better than 3 ppm. The data-dependent MS/MS events were acquired for the five most intense ions detected by full-scan MS, from the *m*/*z* range of 100–1700, above an absolute threshold of 1000 counts and a retention time (RT) threshold of 0.01 min. Selected precursor ions were fragmented at a fixed collision energy of 20 eV with an isolation window of 1.3 amu. To avoid contaminants interferences and technical issues, ions with following *m*/*z* were excluded from MS/MS fragmentation: 121.9906, 228.1955, 279.0944, 338.3414, 622.0290 and 922.0098.

### 2.7. Network Metabolomic Analyses

MSconvert software belonging to Proteowizard package [39] was used for data conversion from Agilent constructor format to mzML files. Data pretreatment was performed with MZMine v. 2.53 software [40] with following parameters: mass detection noise level was set to 1000 for both MS and MS/MS; chromatogram building was achieved with the automated data analysis pipeline (ADAP) chromatogram builder algorithm [41], with a minimum of 3 scans per group, a minimal intensity of 10,000 and *m*/*z* tolerance at 0.005 (or 10 ppm). Chromatogram deconvolution was achieved with ADAP wavelets algorithm [41] with the following parameters: signal/noise threshold 20, minimum feature height 10,000, coefficient threshold 10, peak duration range between 0.00 and 1, RT wavelet range from 0.02 to 0.09, *m*/*z* center calculated as median, *m*/*z* range for MS/MS 0.1 and RT range for MS/MS 0.2. Feature lists were deisotoped with isotopic peaks grouper algorithm with an *m*/*z* tolerance at 0.005 or 10 ppm, and a RT tolerance at 0.1 min. Peak alignment was performed using the join aligner method with *m*/*z* tolerance at 0.005 or 10 ppm, *m*/*z* weight set at 80% and RT tolerance at 0.1 min with a weight of 20%. Aligned chromatograms metadata were used for statistical analyses first with all MS features. Then, MS/MS data were then filtered, and chromatograms and metadata with only MS/MS data were exported to GNPS-FBMN compatible mgf and csv files. Files were processed with MetGem 1.3.6 software [35,42] (https://metgem.github.io/, accessed on 9 March 2022) with following parameters: *m*/*z* tolerance was set at 0.04 and a minimum of two matched peaks was set for similarity between two nodes. All peaks in a +/−17 Da window around the *m*/*z* precursor were removed. Only peaks above 1% of the highest or the six highest peaks in a +/−17 Da window throughout the spectrum were kept. *t*-SNE visualization was constructed with following parameters: at least 1 cosine score above 0.75 was requested. Number of iterations, perplexity, learning rate and early exaggeration were set to 10,000, 6, 10 and 12, respectively. To cope with the high number of MS/MS features, Barnes–Hut approximation angle was used and set at 0.7. Annotations were performed with GNPS (https://gnps.ucsd.edu/ProteoSAFe/libraries.jsp, accessed on 19 October 2022) and MS-DIAL (http://prime.psc.riken.jp/compms/msdial/main.html#MSP, accessed on 19 October 2022) public standards databases.

### 2.8. Glutathione Transferase (GST) Thermal Stability Assay

Fungal GSTs are intracellular enzymes involved in detoxification by white-rot fungi [15]. Interactions between fungal GSTs and extractives can be used as an indicator of wood durability [43]. Thus, the interaction potential of the metabolites found in wood blocks or in mycelia was investigated with ThermalShift Assay (TSA). TSA is based on the capacity of GSTs to bind SYPRO orange stain only once they are denaturated, thus exposing a binding site. Once SYPRO is bound to the GST, fluorescence is emitted, which is used as an indicator of GST denaturation. GSTO2S isolated from Tver cultures, known for its interaction capacity with resveratrol and flavonoids [44], was used. For each of the 60 samples described above (wood and mycelium extract), 11 µL of Tris-HCl 150 mM pH 8.0 buffer, 5 µL of ultra-pure water (Millipore ©, Burlington, MA, USA, Merck™, Darmstadt, Germany); 5 µL of GSTO2S purified from *Trametes versicolor* as described in [45], a at final concentration of 20 µM; 2 µL of SYPRO orange diluted at 1.25%; and 2 µL of extract at initial concentration of 10 mg/mL in DMSO (or 2µL of DMSO only for control wells) were added in one well of a 96-well microplate, on ice and in triplicate. Microplates were spun at 2000 rpm and placed into a ThermalCycler CFX Opus 96 (Bio-Rad ©, Hercules, CA, USA). ThermalCycler temperature was controlled by BioRad CFX Maestro software and set to increase from 5 °C to 95 °C at 5 °C/min in 2 h. Fluorescence was measured continuously at excitation 485 nm and emission 530 nm with HEX fluorochrome. The denaturation temperature, corresponding to the temperature where 50% of the highest fluorescent is measured, was obtained by calculating the derivate of the curve of fluorescence as a function of the temperature. The temperature where the most extreme value (a negative value) of this derivate was achieved corresponds to the denaturation temperature. Denaturation temperatures were then expressed as a thermal shift (ΔT) corresponding to the difference between the observed denaturation temperature and the control denaturation temperature (with DMSO only).

### 2.9. Statistical Analyses

Data analyses were performed using R v. 2022.02.3 software (PBC, https://www.r-project.org/, accessed on 9 March 2022). The Shapiro–Wilk test was used for a data normality check when relevant. Kruskal–Wallis with the Nemenyi Tukey multi-paired comparison test was used for nonparametric tests and Student Newman–Keuls (SNK-test) with Tukey multi-paired comparison test was used for parametric tests. Significant differences are indicated by *** for *p*-values < 0.001, ** for *p*-values < 0.005 and * for *p*-values < 0.05. Principal component analysis (PCA) on the MS and MS/MS features obtained after MZMine v. 2.53 pretreatment was conducted with R and Factoshiny package, and heatmaps were constructed with R.

## 3. Results

### 3.1. Mass Loss and Mycelium Growth

Two Fmed strains (phco36 and LR124) and one Tver strain (BRFM1532) were grown for 4 and 7 months on malt-agar (MA) or on MA with either grapevine ‘Riesling’, oak or beech wood blocks. No morphological differences between Fmed mycelia depending on their substrates were observed (Figure 1). Fmed mycelia were yellowish to brownish and cottony. Tver mycelia were white and less voluminous than Fmed mycelia. Fungal biomass was collected after 7 months of culture, separated from wood blocks, lyophilized, and dry weighed (Figure 2). With grapevine wood, Fmed LR124 produced more biomass than with oak (*p*-value 0.016), and more than Tver BRFM1532 with grapevine wood (*p*-value 0.016). Noticeably, a high variability was observed between replicates. Fmed strain LR124 produced more fungal biomass with grapevine wood than with beech or oak wood (*p*-value 0.016) and compared to Tver with grapevine (*p*-value 0.016), but this was not the case for the strain phco36. 

Individual length, width, height and initial dry weight of all wood blocks were used to calculate dry wood initial density. After seven months of exposure to fungal culture, wood blocks were sampled, separated externally from mycelium, dried for 48 h at 105 °C and weighed to calculate wood mass loss in percent of initial dry weight (Figure 3). Regarding oak wood mass losses, two groups were distinguished for all strains: one group with a high mass loss (“D+”: 54 to 64%) and one group with a low mass loss (“D−“: 9 to 16%). Interestingly, the two groups corresponded, respectively, to wood blocks with a lower initial density (<0.60) and a higher initial density (>0.60) (Appendix A). This was not observed for the two other wood species, despite various initial densities, ranging from 0.53 to 0.84 for grapevine wood and from 0.59 to 0.75 for beech wood blocks (Appendix A). 

Parallelly, mass losses for grapevine wood blocks exposed to Fmed LR124 and Tver showed high variability, contrary to Fmed phco36, with no relation to initial wood density (Figure 3 and Appendix A). After exposure to Fmed phco36, the highest mass loss was observed with oak wood (group “D+”: 56%), then with grapevine wood (43%) and finally beech wood (36%). After exposure to Fmed LR124, mass losses were similar for all wood species (44 to 56%). After exposure to Tver, the highest mass loss was observed on oak wood (group “D+”: 64%) and was similar to those on oak wood after exposure to both Fmed strains. Beech wood mass loss was higher than grapevine wood mass loss after exposure to Tver, and higher than after exposure to Fmed phco36. On the contrary, grapevine wood mass loss was lower than after exposure to Fmed phco36. 

### 3.2. Wood Fiber Analyses

Initial wood blocks contents in lignin, hemicelluloses, cellulose and soluble compounds were compared for each wood species (Figure 4). A similar amount of lignin (20 to 21%) was observed in all wood species. Grapevine wood contained, respectively, 34, 27 and 18% of cellulose, hemicelluloses and soluble compounds, whereas beech wood contained 45, 27 and 8% of each, and oak wood 42, 22 and 15%. Variations in wood compounds contents after 7 months of culture were expressed in final/initial content ratio (Figure 5). A ratio of one means that no changes in compound content occurred, while a ratio higher than one means an increase occurred. After 7 months of exposure with the two Fmed strains, for all wood species, the proportions of each structural compound (hemicelluloses, cellulose and lignin) were similar to initial proportions. Regarding the soluble compound content, it increased in grapevine wood after exposure to the two Fmed strains and in beech wood after exposure to Fmed LR124 (ratios = 1.5 to 1.7). After exposure to all strains, a decrease in soluble compounds content was observed in the less degraded oak wood blocks (ratios = 0.6 to 0.7). 

A similar behavior for structural compounds was observed for Tver BRFM1532 on grapevine and on oak wood blocks: no differences between each structural compound’s proportions were observed, which is characteristic of a simultaneous pattern. A decrease in lignin content in beech wood was detected (ratio 0.5), which suggests a selective delignification on this wood species with this strain. No differences in soluble compounds contents in grapevine, beech and the more degraded oak wood blocks were measured.

### 3.3. Proteomic Analysis

Fmed strains phco36 and LR124 and Tver strain BRFM1532 secretomes on grapevine, beech and oak were extracted from wood blocks after 4 months of culture in triplicates for proteomic analysis. Respectively, 411, 410 and 374 proteins were found (Appendix A). Among them, respectively, 198, 218 and 217 proteins showed an emPAI higher than 1 and were selected. Among them, 173, 169 and 160 proteins could be identified either as oxidoreductases, Carbohydrate Active enZYmes (CAZYmes) or peptidases, respectively. Relative abundances of Fmed phco36, LR124 and Tver BRFM1532 secretomes proteins on grapevine, beech or oak wood blocks, expressed using NSAF, were compared (Figure 6). 

In Fmed phco36 secretome on grapevine wood, 55% of the selected proteins belong to CAZYmes families, including 34% of glycoside hydrolases (GHs) and 14% of carbohydrate esterases (CEs). Oxidoreductases represent 33% of the selected proteins, among them the most represented are manganese peroxidases (MnPs, 25%) and glucose–methanol–choline oxidases (GMC, 5%). Relative abundances between those protein families are similar for Fmed phco36 secretome on oak wood. On the contrary, in Fmed phco36 secretome on beech wood, more CAZYmes, in particular more GHs (42%), and less oxidoreductases, in particular less MnPs (13%), were observed. Fmed phco36 and Fmed LR124 secretomes on beech and oak woods were similar, whereas on grapevine, relatively less GHs (25%) and more MnPs (41%) were observed in the Fmed LR124 secretome. 

In the Tver secretome on grapevine wood, CAZYmes represented 42%, including 28% of GHs, similarly to Fmed LR124 secretome on grapevine wood. Oxidoreductases represented 45% of the Tver secretome on grapevine wood, but contrary to Fmed secretomes, the most represented oxidoreductases were lignin peroxidases (LiPs), representing 23% of the selected proteins. MnPs were also secreted, representing 9%. Contrary to what was observed with Fmed secretomes, the relative amount of MnPs and LiPs was different in the Tver secretome depending on the substrate. On beech wood, LiPs and MnPs represented similar proportions of the selected proteins (both 8%), whereas on oak wood, they represented 9 and 22%, respectively. 

Relative abundances of each secreted oxidoreductases were compared for each strain on the three different woody substrates (Figure 7). On grapevine and oak wood, nine MnP isoforms were observed in the two Fmed strains secretomes, and only eight on beech wood. On all substrates, the same number of other oxidoreductases (one dye-decolorizing peroxidase—DyP, one laccase, 2 lytic polysaccharide monooxygenases—LPMOs, six glucose–methanol–choline oxidoreductases—GMCs and one quinone oxidoreductase) was observed in the two Fmed strains secretomes. On grapevine wood, the most abundant MnP isoform secreted by both Fmed phco36 and LR124 was MnP2l. The isoforms MnP10l and MnP11l (and MnP5l) were more abundant on grapevine and oak woods than on beech wood. 

The same oxidoreductases isoforms (five LiPs, nine MnPs, two DyPs, two laccases, two versatile peroxidases—VPs, one copper radical oxidase—CRO, two xylose dehydrogenases, five glyoxal oxidases and one glucose oxidase) were observed on Tver BRFM1532 secretomes on the three substrates. On grapevine wood only, the most abundant secreted oxidoreductase is the isoform LiP2. Contrary to Fmed secretomes, no differences between MnP isoforms depending on the substrate were observed.

Relative abundances for each CAZYmes and peptidases were also compared for each fungal strain secretome (Appendix A). None of them was higher or less abundant in both Fmed strains secretomes depending on the substrate. 

### 3.4. Non-Targeted Metabolomic Analysis

Extracts from wood blocks (grapevine, beech, oak), not exposed to fungal degradation (control wood) or after 4 months of exposure to each of the three strains (Fmed phco36, Fmed LR124 or Tver BRFM1532), as well as the mycelia of the three strains grown on the surface of each wood or on malt-agar (control mycelium) were analyzed using HPLC-ESI-HRMS/MS DDA in positive ionization mode. After ADAP chromatogram construction and deconvolution, isotopic peaks grouping and peak alignment, 4190 MS features were selected. Among them, 2812 features were found only in mycelia extracts and 2268 only in wood extracts. 

All obtained MS features were used for principal component analysis (PCA), first with all extracts (from both wood and mycelium samples) with quality control samples (QC) to check the technical analysis quality (Appendix A). QC samples were grouped together, which demonstrates good technical quality. PCA was then conducted on all samples without QC and sample types (mycelium or wood extracts) were statistically different on dimensions 1 (*p*-value 2.7 × 10^−4^) and 3 (*p*-value 2.0 × 10^−8^), which explain, respectively, 11.87 and 7.09% of the total variance (Appendix A). Regarding those dimensions, two samples were far from their replicates: “R124M1” and in a lesser extent “C124M3”, corresponding, respectively, to Fmed LR124 mycelium grown on grapevine wood (replicate no. 1) or on oak wood (replicate no. 3). Those samples will be considered carefully during further analyses. 

PCA analyses were then conducted separately on wood samples only or on mycelium samples only. Among the wood samples, wood species were significantly different on dimensions 2 (*p*-value 4.7 × 10^−2^) and 3 (*p*-value 1.2 × 10^−3^), explaining, respectively, 12.4 and 9.1% of the total variance between wood samples (Appendix A). Noticeably, a higher heterogeneity between oak and beech wood samples than between grapevine wood samples was observed. 

Among the grapevine wood samples only (Figure 8a), statistical differences were observed between fungal species (*p*-value 2.6 × 10^−5^) and strains (*p*-value 1.2 × 10^−4^) on dimension 2, which explains 19.7% of total grapevine wood samples variance. Figure 8a shows that this effect mainly separates the control wood from wood exposed to any of the three fungal strains or two fungal species. Similar effects were observed for oak wood samples (*p*-values, respectively, 3.6 × 10^−4^ and 1.3 × 10^−3^) on dimension 3, which explains 12.66% of total oak wood samples variance (Figure 8b), and for beech wood samples regarding dimension 4 (Figure 8c). Among all grapevine wood samples, 588 of the total selected MS features were found, 69 of the latter were found only in control wood and 7 were found in all samples after exposure to fungal strains, but not in control wood (Figure 9). Consistent with PCA analysis, hierarchical clustering separated control wood from wood exposed to fungal species, and few differences between grapevine wood samples after exposure to each fungal strain or species were observed. 

PCA analysis on mycelium samples showed statistical differences among fungal species and strains on dimensions 1 (*p*-values, respectively, 2.9 × 10^−2^ and 1.1 × 10^−2^) and 2 (*p*-values, respectively, 1.1 × 10^−6^ and 8.0 × 10^−6^), which explain, respectively, 17.5 and 11.3 % of total mycelium samples variance (Appendix A). As only Tver BRFM1532 was graphically separated from the two Fmed strains, the two Fmed strains will not be distinguished for further analyses. Noticeably, the sample “R124M1” was far from its replicates, particularly on dimension 1, and should be considered carefully. Interestingly, mycelia were statistically different upon the wood species on which they were grown on (*p*-value 2.3 × 10^−3^) dimension 3, which explains 7.2% of total mycelium samples variance (Appendix A). 

In order to highlight Fmed mycelium metabolomic changes depending on its substrate, relative abundances of Fmed mycelium significantly different metabolites between mycelium grown either with grapevine or beech wood (851 MS features, Appendix A), or with grapevine or oak wood (796 MS features, Appendix A) were represented with hierarchical clustering. As observed with PCA analyses, a high heterogeneity between replicates was observed. 

### 3.5. Molecular Networking

Molecular networking, using MS/MS data, is a non-targeted method enabling features annotation with relative accuracy. During DDA analysis, each 0.01 min, the five most intense MS peaks were fragmented. Thus, MS/MS data are available only for some of the previously selected MS features. Consequently, the latter were filtered to keep only features with MS/MS data. This led to 3011 MS/MS features (Appendix A), among which, 1000 were found only in wood samples, 1245 only in mycelium samples and 760 in both. 

The 3 011 MS/MS features were mapped with the *t*-SNE algorithm (Figure 10). In *t*-SNE graphical representation, each feature is represented by a “node”. The less distance there is between each node, the more similarities there is between MS/MS data of the corresponding features, and the more the latter are prone to share chemical structural characteristics. Nodes were gathered into 91 clusters with the MetGem clustering algorithm set with a minimal of five features per cluster. As *t*-SNE algorithm has the particularity to keep distances between all nodes even between different clusters, it should be kept in mind that close clusters might gather features with some common fragment peaks or neutral loss. Annotation was achieved according to free GNPS and MS-DIAL databases, and cosine score (cs) was given as a confidence indicator. The closer the cs is to one, the more similarities there are between databases standards and experimental features.

#### 3.5.1. Wood Samples Metabolome

As shown with descriptive statistical analysis (PCA and hierarchical clustering), the main differences between all samples were attributable to sample type, i.e., their woody or mycelium origin. Thus, wood samples and mycelium samples compositions will be described separately. 

Among wood samples, wood species was the most discriminant factor (see Section 3.4). Wood species specific MS/MS features, i.e., all MS/MS features found only in one of each wood species, are listed in Appendix A. Appendix A are extracted from Appendix A to highlight specificities among samples. Among the 211 grapevine wood specific features, 37 were found only in control wood, and 137 only after exposure to fungal degradation (Appendix A).

Differences between grapevine wood samples after exposure to fungal degradation are detailed in Appendix A (which includes features found in other wood species samples). In grapevine control wood only (Appendix A), six features were mapped in clusters 11 to 14. Two of them were annotated with an in silico database (ISDB) as “compound 1”, with a structure close to *epsilon*-viniferin (cs 0.55 and 0.60, *m*/*z* 453.133, cluster 14, Appendix A), and two others either as s-viniferin (cs 0.78, *m*/*z* 455.149, cluster 13) or as an analog of the latter. Other features found only in grapevine control wood were annotated as coniferaldehyde (cs 0.94, *m*/*z* 179.070, cluster 75), epicatechin (cs 0.93, *m*/*z* 291.086, noise), 3,4,5-trimethoxyphenol (cs 0.93, *m*/*z* 185.081, noise), syringaldehyde (cs 0.93, *m*/*z* 183.065, cluster 65) or 4-hydroxybenzaldehyde (cs 0.91, *m*/*z* 123.044, cluster 65). 

In control grapevine wood and in grapevine wood exposed to Fmed only (i.e., not after exposure to Tver), 14 features were detected (Appendix A), among them 4 only found in grapevine wood (Appendix A). The latter were annotated among others as betulinic acid (cs 0.87, *m*/*z* 439.357, noise) and resveratrol (cs 0.86, *m*/*z* 229.086, cluster 71). Among features not specific to grapevine wood, two were annotated as glycerol-1-stearate (cs 0.94, *m*/*z* 359.316, cluster 10) and tripropylene glycol butyl ether (cs 0.86, *m*/*z* 249.206, cluster 35). Those features might have been degraded by Tver and not by Fmed. Conversely, six features were in control grapevine wood and after exposure to Tver, but not after exposure to the two Fmed strains, suggesting that they were degraded by Fmed and not by Tver (Appendix A). Among them, two were found only in grapevine wood and were annotated as sumaresinolic acid (cs 0.62, *m*/*z* 455.352, noise) or as an analog to the compound KU036-11-7 (cluster 8, see Appendix A). The others, not specific to grapevine wood, were namely annotated as analogs to a polyketide (cluster 30, see Appendix A) or to the monoterpene neryl-acetate (cluster 33, see Appendix A). 

Features found in grapevine wood exposed to fungal degradation, and not in control grapevine wood, might be fungal metabolites or wood degradation products. Among them, 125 features were specific to Fmed on grapevine wood, i.e., found in grapevine wood samples only after exposure to Fmed, and not after exposure to Tver (Appendix A). Among them, 74 were found only in grapevine wood (Appendix A). They were annotated as ceramides (cluster 4), stilbenoids (clusters 11 to 15), an acetophenone (cluster 44), 3-methylglutaconic acid (cluster 58), probably other phenolic compounds (cluster 61), 7,8-dihydroxy-4-methylcoumarin (cluster 75) hydroxybenzoylcholine (cluster 84), isoeleutherin and choline (cluster 89), or as analogs to diacylglycerol trimethylhomoserine (cluster 56). 

Conversely, 95 features were found in grapevine wood exposed to Tver and not exposed to Fmed (Appendix A). Among them, 23 were found only in grapevine wood (Appendix A). They were annotated as stilbene oligomers (clusters 14 and 15), pyrogallol (cluster 51), 13-keto-9Z,11E-octadecadienoic acid (cluster 60), gameXpeptide (cluster 66), loliolide (cluster 72), labdanediol (cluster 81) or methyl asterrate (cluster 86). Interestingly, few of them probably share common chemical structure with features specific to grapevine wood exposed to degradation by Fmed (belonging to clusters 14 and 15 namely). On the contrary, most of them probably belong to different chemical classes, as they were mapped in distinct clusters. 

#### 3.5.2. Mycelium Samples Metabolome

Considering mycelium samples, fungal species were the most discriminant factor, and differences among wood species were also observed (see Section 3.4). Among the 2 005 MS/MS features found in mycelium samples, 541 were only in Fmed mycelium, and 418 only in Tver mycelium (Appendix A).

MS/MS features found in Fmed mycelium only are listed in Appendix A. Some of them (227 features) were found only in Fmed mycelium grown with grapevine wood, only with beech wood (213 features) or only with oak wood (70 features). However, a high variability between individuals was observed. Only one feature, which was not annotated with the databases, was found in all individual Fmed mycelia grown with beech wood (*m*/*z* 491.097, cluster 16, RT 6.95 min). Regarding mycelia grown on grapevine wood, one sample (“R124M1”), accordingly to PCA analysis, was markedly different from the others, containing more features. Among the 227 features found only in Fmed mycelia grown on grapevine wood, 20 were found in at least two individuals (Appendix A). The latter were annotated as *allo*-protolichesterinic acid (cs 0.68, *m*/*z* 330.264, cluster 83, Appendix A) or analog (cluster 69), and as analogs to diacylglyceryl-trimethylhomoserine (DGTS, cluster 54) or 3-methylglutaconic acid (cluster 58) (Appendix A).

Interestingly, but only in one individual (“R124M2”, replicate no. 2 of Fmed LR124 mycelia grown with grapevine wood), 14 of them were mapped in cluster 12 to 15 and annotated as stilbene trimers (probably four isomers, cs 0.50 to 0.64, *m*/*z* 681.212, cluster 15, RT 7.0, 7.9, 8.1 and 8.4 min), *epsilon*-viniferin (probably two isomers, cs 0.90 and 0.92, *m*/*z* 455.149, cluster 13, RT 7.5 and 8.0 min), or in silico annotated as stilbene dimers (probably three isomers, close to viniferin, see Appendix A). In Tver mycelium grown on grapevine wood, nine features were mapped in clusters 11 to 15 as well and annotated among others as *epsilon*-viniferin (two isomers), *s*-viniferin or as stilbene dimer (in silico annotated).

Common MS/MS features found in different relative abundances in Fmed and Tver mycelia grown with grapevine wood are listed in Appendix A. All features more abundant in Fmed mycelium (21 features) were found in all other Fmed mycelia as well, and not in Tver control mycelium, suggesting that they are typical Fmed features, but not specific to the presence of grapevine wood. They were among others annotated as betaine (cs 0.99, *m*/*z* 118.086, cluster 29), 13-keto-9Z,11E-octadecadienoic acid (cs 0.74, *m*/*z* 295.227, cluster 60) or *allo*-protolichesterinic acid (cs 0.61, *m*/*z* 325.274, cluster 76).

Conversely, some of the common features that were more abundant in Tver mycelium grown on grapevine wood than on Fmed mycelium grown on grapevine wood (121 features) were not found in Tver control mycelium (56 features), suggesting metabolomic changes in Tver mycelium due to the presence of wood. All those features were found in the presence of all wood species except four of them, found only in the presence of grapevine or oak wood. The latter were annotated as *epsilon*-viniferin (cs 0.91, *m*/*z* 455.149, cluster 13), or as analogs to hepta- and octapropylene glycol (cs 0.57, 0.64 and 0.63, *m*/*z* 493.852, 551.894, 580.915, cluster 2). 

Finally, MS/MS features found in different relative abundances in Fmed mycelium grown with grapevine wood compared to Fmed mycelium grown with the two other wood species are listed in Appendix A. Among them, 12 were more abundant in the presence of grapevine wood compared to the two other wood species. They were annotated as *allo*-protolichesterinic acid (cs 0.61, *m*/*z* 325.274, cluster 76) and *cis*-5,8,11-eicosatrienoic acid (cs 0.92, *m*/*z* 307.263, cluster 83), or as analogs to 9Z,11E,13E-octadecatrienoic acid methyl ester (cluster 31), 9-oxo-10E,12Z-octadecadienoic acid (cluster 41), sulfonolipid 11:0;O/22:5 (cluster 49) or 3-keto-9Z,11E-octadecadienoic acid. Conversely, 12 were less abundant in the presence of grapevine wood, and were annotated as hexosylceramides (HexCer, cluster 3), 1-(2-hydroxy-4,6-dimethoxyphenyl)-2-methoxyethan-1-one (cluster 86), 13S-hydroxy-9Z,11E,15Z-octadecatrienoic acid (cluster 83), or as analogs to a steroid compound (see Appendix A, cluster 21), phytosphingosine (cluster 47), octa- or decapropylene glycol (cluster 55). 

Thus, among the 3 011 MS/MS features in all samples, a very few of them were observed only with specific fungal wood associations, in particular to Fmed and grapevine wood: six features were found in control grapevine wood and not after exposure to Fmed, suggesting a specific degradation by Fmed (sumaresinolic acid, analogs to KU036-11-7, to a polyketide and to neryl acetate), while others (125 features) were found after exposure to Fmed but not in control, in other wood species or after exposure to Tver (ceramides, glycerolipids, stilbenoids, phenolic compounds, other aromatic compounds, monoterpenoids and one alkenyl alcohol). However, a particularly high heterogeneity in wood samples metabolomic composition was observed. This was also true for mycelium samples, in which 227 features were specific to Fmed mycelium grown with grapevine wood, but none of them were found in all replicates. Within Fmed mycelia, 24 features were highlighted as either more abundant in the presence of grapevine wood (including *allo*-protolichesterinic acid, cis-5,8,11-eicosatrienoic acid (cs 0.92, *m*/*z* 307.263, cluster 83), or analogs to fatty acids or sphingolipid), or less abundant (hexosylceramides, an acetophenone, a fatty acid, or analogs to a steroid compound, phytosphingosine or propylene glycols).

### 3.6. Thermal Stability Assay (TSA)

Changes in metabolomic composition within wood blocks or mycelium could be evaluated using TSA. Thermal shift (ΔT), i.e., the difference between GSTO2S denaturation temperature in the presence of the extract compared to its control denaturation temperature, were obtained for each extract (Figure 11). St Martin wood extract was used as a positive control. An increase in the GSTO2S denaturation temperature (+4 to 11 degrees Celsius (°C)) was observed in the presence of Fmed LR124 mycelium grown on beech and on oak, and in the presence of all grapevine wood extracts. This increase reveals a stabilization of GTSO2S in the presence of the extracts. The stabilization was higher with extracts from grapevine wood exposed to fungal growth (+8 to 11 °C) than with those from control grapevine wood (+6 °C). 

## 4. Discussion

Physiological analyses highlighted Fmed traits that are common on the three woody substrates: the two Fmed strains are able to degrade the three wood species, with degradation rates ranging from 43 to 56% *w*/*w* dw. Moreover, the two Fmed strains show on all wood species similar morphologies and are associated with a loss of all wood structural compounds in similar proportions, which is characteristic of a simultaneous degradation pattern [46], consistent with previous observations [15,47]. Tver shows a simultaneous degradation pattern on grapevine and on oak wood as well but seems to act as a selective wood degrader on beech wood. Such regulation of a white-rot fungi degradation pattern can be due to substrate properties, such as its structural composition or its nitrogen content [46]. One possible explanation is a regulation of the fungal secretome upon the substrate. For example, cellulases secreted by white-rot Basidiomycota can be regulated upon nitrogen source or carbohydrates and lignin contents [48]. Surprisingly, oxidoreductases, which are mainly involved in lignin mineralization, were less represented in the Tver secretome on beech compared to those on grapevine and oak wood. Moreover, the two Fmed strains secretomes show less oxidoreductases on beech than on grapevine or on oak wood, despite no differences in global mass loss or wood degradation pattern (simultaneous on the three substrates). Considering the fact that initial wood lignocellulosic compositions were relatively similar, such observations suggest specific activities for Fmed secreted wood-degrading enzymes. Indeed, wood-degrading enzymes, in particular class-II peroxidases (which gather MnPs, LiPs, VPs and GPs), can be better adapted to different lignin or hemicelluloses compositions [47,49].

Fmed secretomes show few differences upon the substrate. The most noticeable regulation concerns MnPs, which are in relatively lower abundancy on beech wood compared to oak and grapevine wood. More interestingly, the higher abundance of the isoform MnP2l compared to the other isoforms in both Fmed strains secretomes only on grapevine wood suggests a specific regulation for this isoform on grapevine wood. MnPs isoform regulation has already been reported in Tver or in the other white-rot *Phanerochaete chryosporium* secretomes depending on their growth conditions [50]. Such observations were suggested to explain white-rot fungi gene multiplicity, despite the regulation mechanisms are still not fully understood.

Noticeably, with all fungal strains, two groups were distinguished between oak wood blocks: one with low initial density (lower than 0.60) and with the highest mass loss among all wood species, compared to the other with the highest initial densities (above 0.60), which gather wood blocks not degraded after exposure to any of the three fungal strains. Indeed, previous studies showed that wood density, depending on wood and fungal species, is involved in wood durability [43]. Moreover, grapevine wood initial densities ranged between 0.53 and 0.84 with no effect on degradation rate. Thus, our results suggest that grapevine resistance to fungal decay by Fmed (or Tver) is not related to its density.

Fmed strain LR124 produces more biomass on grapevine wood than on oak wood, and more than Tver strain BRFM1532 on grapevine wood. Mass losses after exposure to Fmed strain phco36 were higher for grapevine wood than for beech wood, and than for grapevine wood after exposure to Tver strain BRFM1532. Parallelly, wood mass losses after exposure to Tver BRFM1532 were higher for beech wood blocks than for grapevine. Our results suggest that Tver BRFM1532 wood degradation capacity is higher on beech wood than on grapevine wood. Such observations could be due to fungal adaptation to wood chemical and anatomical traits [51]. However, intraspecific variability between the two Fmed strains is observed. Thus, the capacity to degrade more grapevine wood compared to other wood species might not be a specific Fmed trait. Moreover, a high heterogeneity between wood samples regarding degradation rates is observed, particularly between grapevine wood samples. This might be due either to fungal variability or to initial grapevine wood blocks heterogeneity. This would be consistent with the fact that grapevine is particularly exposed to biotic and abiotic stresses, such as pruning or microbial colonization, which could have led locally to physical and chemical changes even between close woody tissues [52,53]. 

Wood samples show different metabolomic compositions depending on the wood species. This is particularly highlighted with *t*-SNE graphical representation, where features only found in grapevine, beech or oak woods appear in distinct clusters. Features typical to grapevine wood are annotated as stilbenic oligomers or as resveratrol. Annotated stilbene oligomers include dimers, trimers and probably tetramers (*m*/*z* 908.279, clusters 11 and 12). Those observations are consistent with the known stilbene abundances in grapevine wood [54,55]. Some stilbenes, such as resveratrol, are found both in control wood and after exposure to fungal degradation, suggesting that not all of them were degraded during culture, and that they have no or a low fungitoxicity. This is also the case for flavonoids, such as one analog to epicatechin, not detected anymore after exposure to fungal degradation. Other compounds, found only in grapevine wood after exposure to fungal degradation, are annotated as stilbenes and show structural differences compared to those found in non-degraded wood. In particular, oxyresveratrol, an oxidized form of resveratrol, was found only in grapevine wood after exposure to fungal degradation and suggests an oxidation of resveratrol during the culture. Features clustered with stilbene dimers and trimers with *m*/*z* 907.275 and 908.279 (cluster 15), annotated as stilbene tetramers, could result from a stilbene oligomerization during fungal culture. One *epsilon*-viniferin isomer, different from the one found in non-degraded wood, also suggests that isomerization occurred during the culture. Stilbene oxidation, followed by oligomerization, can occur either spontaneously or under fungal activity [21,56,57,58]. Chemical mechanisms of stilbene oligomerization were shown to involve peroxidases [16,17,18] and laccases [19], produced for example by *Trametes pubescens*. The latter can catalyze hydroxystilbenes oxidation and dimerization [21]. This is consistent with the presence of peroxidases and laccases in Fmed and Tver secretomes. Stilbenes oxidation and oligomerization could play a role either in detoxication processes or as a supply in additional carbon source [59,60]. 

Potential biological activities of the samples were investigated through their interactions with a fungal GST (Tver GSTO2S). TSA is a tool to compare the potential of interaction between wood or mycelium extracts and a fungal glutathione transferase (GST). Interactions between extractives and GSTs can be used as an indicator of extractives biological properties [43,61]. Grapevine wood samples after exposure to fungal degradation (with the three strains) interact more with the GST than control grapevine wood, which might reflect metabolomic changes, affecting biological properties of the wood extracts. Those changes can be related to the changes observed in stilbenoid composition in grapevine wood samples. Indeed, stilbene oligomers, shown to be accumulated in response to fungal degradation, are known to exhibit higher fungitoxicity than their respective monomers [62]. As white-rot fungi can oxidize and oligomerize stilbenoids, oligomerization might be a step in the detoxification process involving GSTs [14]. Oligomerization also affects stilbenoid chemical properties, such as their solubility, which could be involved in their biological properties [54,63,64]. Such metabolomic changes due to fungal exposure might contribute to grapevine wood colonization by both Fmed and Tver. 

Regarding metabolomic changes in grapevine wood, a few features found in grapevine control wood and after exposure to Tver are not found after exposure to the two Fmed strains. They are annotated as sumaresinolic acid, or as analogs to KU036-11-7 (Appendix A), to a polyketide or to neryl acetate. They could have been degraded or mineralized by Fmed only. Sumaresinolic acid is a triterpenoid that can be found in wood, such as in *Coffea canephora* trunk [65]. Terpenoids were not previously reported in grapevine wood metabolomic studies to our knowledge, which may be due to the high prevalence of phenolic compounds, including stilbenes and flavonoids [10,11,66]. Compound KU-036-11-7 is a triterpenoid with two aromatic rings identified in *Eremophila rugosa* leaves [67]. MS/MS data suggest that the feature found in our samples, with *m*/*z* 491.373 and annotated as an analog of KU036-11-7, share chemical structural parts with KU036-11-7. It is also closely related to the other features found in clusters 7 and 8, which include namely one glycosylated phenolic ester. Glycosylation of phenolic compounds plays several essential roles in plants, including increasing fungal growth inhibition activities [68]. The degradation of glycosylated compounds by Fmed could be a step in its grapevine wood degradation mechanisms. 

On the contrary, features found only in grapevine wood after exposure to Fmed and not in any other wood samples (121 features) suggest that Fmed regulates its metabolism on grapevine wood. Those features might be wood degradation products or fungal metabolites, and are annotated as ceramides, monoglycerolipids, stilbenoids, phenolic compounds, other aromatic compounds, monoterpenoids or alkenyl alcohols. Ceramides and glycerolipids are members of the sphingolipids, found in both plants and fungi [69]. Their structural specificities can help to decipher their fungal or plant origin [70]. Ceramides found in our samples contain the sphingoid base 4-hydroxysphinganine, or phytosphingosine. The latter is found both in plants and fungi and is the sphingoid base of glycosyl inositol phosphoryl ceramides, which are typical membrane lipids. Thus, they can result from both wood or fungal membrane degradation, either due to wood degradation or to fungal autolyse. Glycerolipids can be found in fungi mostly as triglycerides, and often play a role as storage compounds [71]. However, living tissues extraction procedure might release lipases, often still active in organic solvents, which could degrade fungal or wood lipids, and artificially produce smaller lipids such as monoglycerids or fatty acids [72]. To further investigate Fmed specific lipids, the extraction procedure could be improved specifically for lipids, namely through solvent acidification. Here, it should be noted that lipids might be overestimated compared to carbohydrates as they are better ionized in MS [73]. Extraction procedures could also be improved to investigate more specifically wood extractives or fungal metabolites belonging to different chemical families. Moreover, heterogeneity between wood samples metabolomic composition is particularly high, and thus it is difficult to conclude the fungal-substrate specificities.

In mycelium samples, 227 features are found only in Fmed mycelia grown with grapevine wood. However, a high heterogeneity between samples is observed. Among them, 20 are found in at least two replicates, and are annotated as *allo*-protolichesterinic acid, or as analogs to a nitrogenous aromatic compound (cluster 30), to DGTS (cluster 54) or to a dicarboxylic acid (cluster 58). A few other features only found in Fmed mycelium and more abundant when grown on grapevine wood (12 features) are annotated as *allo*-protolichesterinic acid, *cis*-5,8,11-eicosatrienoic acid (cluster 83), and analogs to fatty acids or to sphingolipids. Finally, 21 features are found in both Fmed and Tver mycelium and more abundant in Fmed than in Tver mycelium grown on grapevine wood and are annotated as betaine, 13-Keto-9Z,11E-octadecadienoic acid or *allo*-protolichesterinic acid. The latter is a gamma-lactone commonly found in lichens with antimicrobial activities [74,75]. No previous study reporting its production by fungi was found, and further experiments are needed to validate its annotation. Features annotated as fatty acids or analogs are stearic acid (C18) derivatives, which are consistent with the typical fungal sphingolipids structure [69]. They might be present as free fatty acids in Fmed mycelium, or as a result of enzymatic lipids degradation during extraction [71,72]. 

## 5. Conclusions

Fungal-substrate specificities were studied through physiological and metabolomic traits during grapevine, beech or oak wood block colonization by two Fmed strains and the white-rot model Tver. Fmed was able to degrade the three wood species with similar degradation rates and showed a simultaneous degradation pattern on all of them. A noticeable intraspecific variability was observed between the two Fmed strains regarding wood mass loss and fungal biomass after 7 months of culture, as well as a high heterogeneity between samples. The latter highlighted both fungal *inocula* and grapevine wood samples heterogeneity. Moreover, the highest degradation rate was observed for oak wood with all fungal strains. For this species only, initial wood density was the most critical factor explaining wood degradation rate, and only low-density oak wood blocks were degraded. Accordingly, Fmed wood degradation mechanisms seem not to be specifically adapted to grapevine wood. However, noticeable traits, such as MnP2l isoform higher abundance on grapevine wood, could be involved in Fmed adaptation in other environmental conditions.

Molecular networking was successfully used to describe samples metabolomic composition. Changes in grapevine wood composition after exposure to the three fungal strains were observed, such as a diminution of resveratrol and an increase of stilbene oligomers. After fungal degradation, grapevine wood extracts also interacted more than non-degraded grapevine wood extracts with the fungal GSTO2S, which suggests differences in the extracts’ biological activities. Specificities between Fmed and grapevine wood were discussed regarding both wood and mycelium metabolites.

This study investigated the role of wood composition as Fmed substrate and described physiological and metabolomic changes due to fungal exposure. However, other factors, biotic or abiotic, probably play a key role in fungal wood colonization and might be involved in Fmed adaptation in vineyards. For example, in response to stress such as pathogen attacks, extractives are de novo synthesized or accumulated in grapevine wood. Thus, Fmed might be particularly adapted to grapevine stress responses. Grapevine wood physiological state, or environmental conditions might also play a role in Fmed traits. Thus, the proposed physiological and non-targeted metabolomic approach explored only one of the multiple and complex factors involved in grapevine wood colonization by Fmed, and should be further used and completed in order to better understand Fmed widespread in vineyards.

## Figures and Tables

**Figure 1 jof-09-00536-f001:**
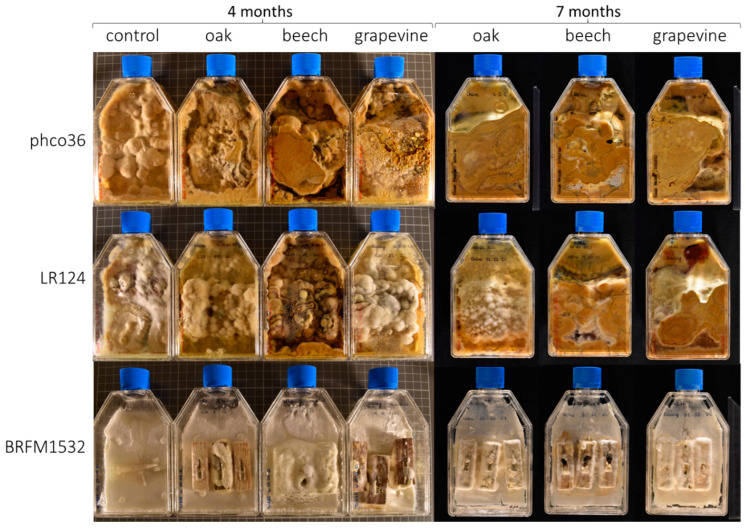
Morphological observation of culture flasks after 4 and 7 months of Fmed phco36, Fmed LR124 and Tver BRFM1532 cultures on malt-agar (control) or on malt-agar with wood blocks (3 blocks per flask) of oak, beech or grapevine wood.

**Figure 2 jof-09-00536-f002:**
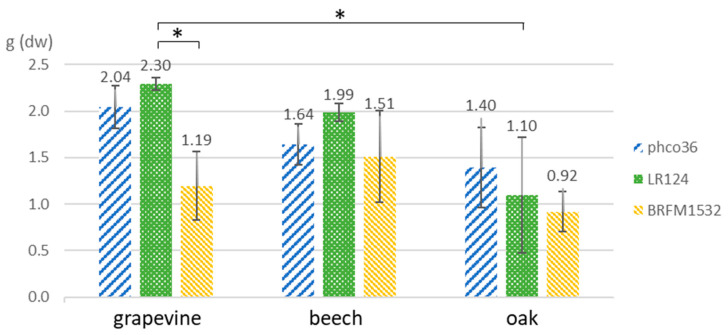
Mycelium dry weight per flask collected after 7 months of culture. Kruskal–Wallis with Nemenyi paired test (*: *p*-value < 0.05).

**Figure 3 jof-09-00536-f003:**
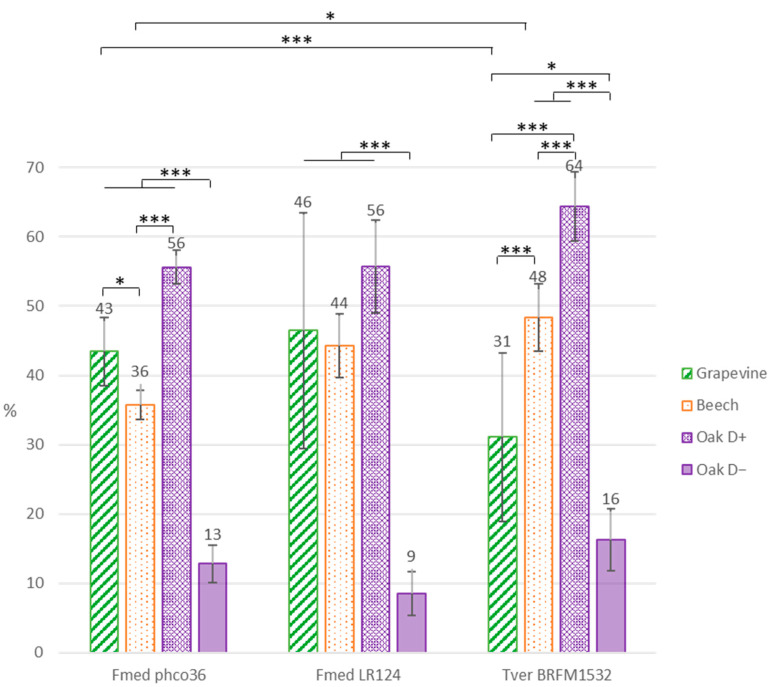
Wood blocks mass loss in percentage of their initial dry weight. Error bars are the standard error of the mean (n = 10 except for oak wood exposed to Fmed: n = 5 and to BRFM1532: n = 4 and 6, respectively, for oak D+ and oak D−). Oak wood mass losses were distinguished into two groups: D+ = showing a higher mass loss, and D− = showing a lower mass loss. Statistic test: Student–Newman–Keuls (***: *p*-value < 0.001; *: *p*-value < 0.05).

**Figure 4 jof-09-00536-f004:**
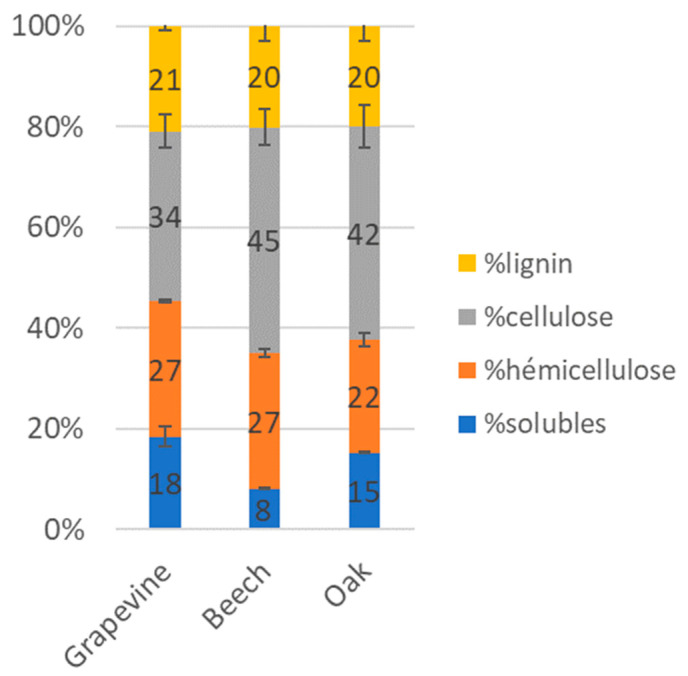
Wood initial proportions in lignin, cellulose, hemicelluloses and soluble compounds for grapevine, beech and oak wood blocks. Error bars give the standard error of the mean between two technical replicates.

**Figure 5 jof-09-00536-f005:**
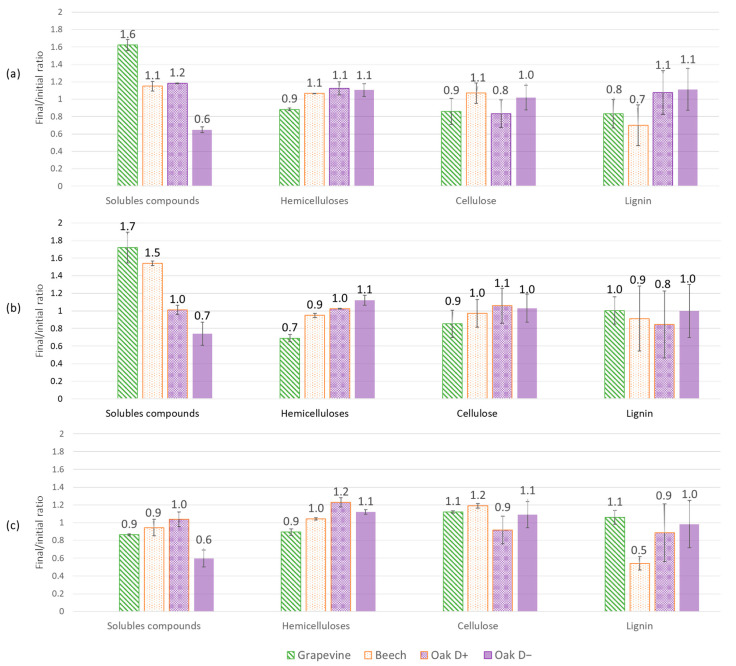
Final/initial contents ratios for each wood compound (solubles, hemicelluloses, cellulose and lignin) after 7 months of culture in grapevine, beech and oak wood blocks with (**a**) Fmed phco36, (**b**) Fmed LR124 and (**c**) Tver BRFM1532. Error bars include standard error of the mean between the two technical replicates after 7 months of culture and for initial composition. Oak wood blocks were separated according to the two groups observed regarding their mass loss: “Oak D+” = higher mass loss; “Oak D−“ = lower mass loss.

**Figure 6 jof-09-00536-f006:**
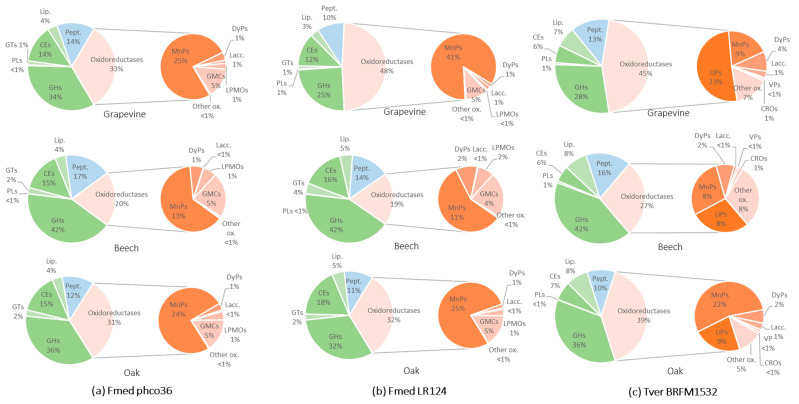
Relative abundances (based on NSAF values) of the main protein families involved in wood degradation found in (**a**) Fmed phco36, (**b**) Fmed LR124 or (**c**) Tver BRFM1532 secretomes in grapevine, beech or oak wood blocks at 4 months of culture. GHs = glycoside hydrolases, PLs = polysaccharide lyases, GTs = glycosyl transferases, CEs = carbohydrate esterases (excluding CEs specifically identified as lipases), Lip. = lipases, Pept. = peptidases, MnPs = manganese peroxidases, DyPs = dye decolorizing peroxidases, Lacc. = laccases, LPMOs = lytic polysaccharide mono-oxygenases, GMCs = glucose-methanol-choline oxidoreductases, Other ox. = other oxidoreductases, LiPs = lignin peroxidases, VPs = versatile peroxidases, CROs = copper radical oxidases.

**Figure 7 jof-09-00536-f007:**
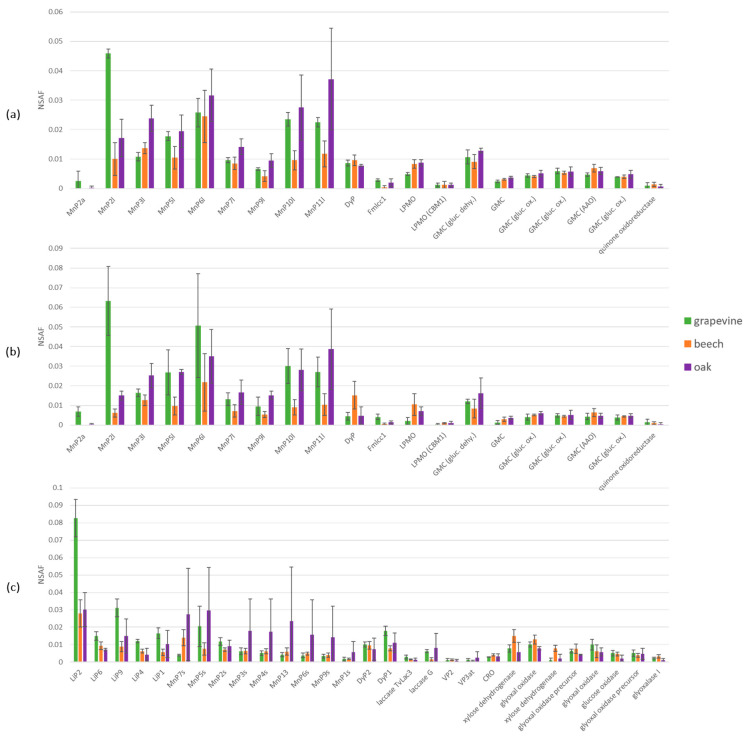
Relative abundances (based on NSAF values) of oxidoreductases secreted by (**a**) Fmed phco36, (**b**) Fmed LR124 or (**c**) Tver BRFM1532 on grapevine, beech or oak wood at 4 months of culture.

**Figure 8 jof-09-00536-f008:**
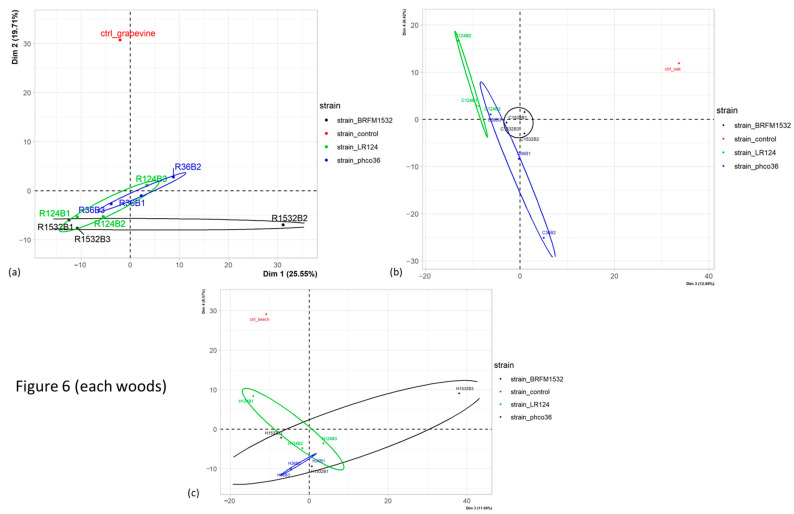
PCA analysis (**a**) on grapevine wood samples (dimensions 1 and 2), or on (**b**) oak and (**c**) beech wood samples (dimensions 3 and 4), after exposure to each fungal strain or control woods. For sample names, see Appendix A caption.

**Figure 9 jof-09-00536-f009:**
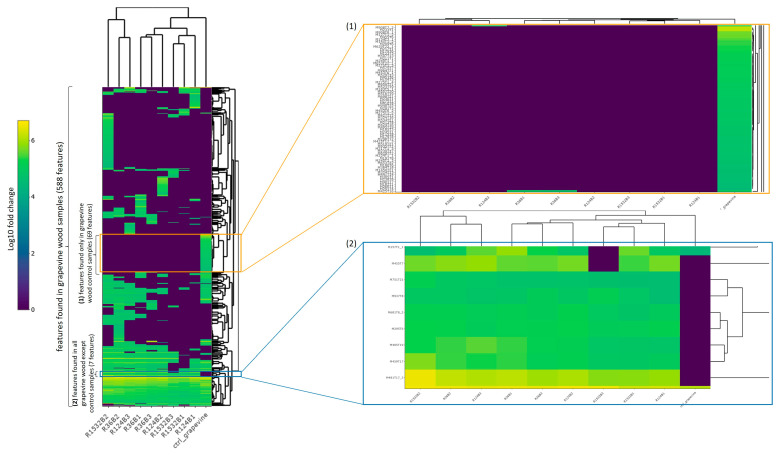
Log10 relative abundances of selected MS features found in grapevine wood samples: among those 588 features, 69 were found only in control wood samples (close-up no. (**1**)), and 7 were found only after exposure to all fungal strains and not in the control grapevine wood (close-up no. (**2**)). Sample names are written as follows: grapevine wood exposed to Fmed phco36 (“R36B*X*”), to Fmed LR124 (“R124B*X*”) or to Tver BRFM1532 (“R1532B*X*”), where “*X*” refers to each of the three biological repetitions, grapevine control wood (“ctrl_grapevine”). Features names (close-up rows) are given according to their *m*/*z* (“M”) and retention time (“T”) in min.

**Figure 10 jof-09-00536-f010:**
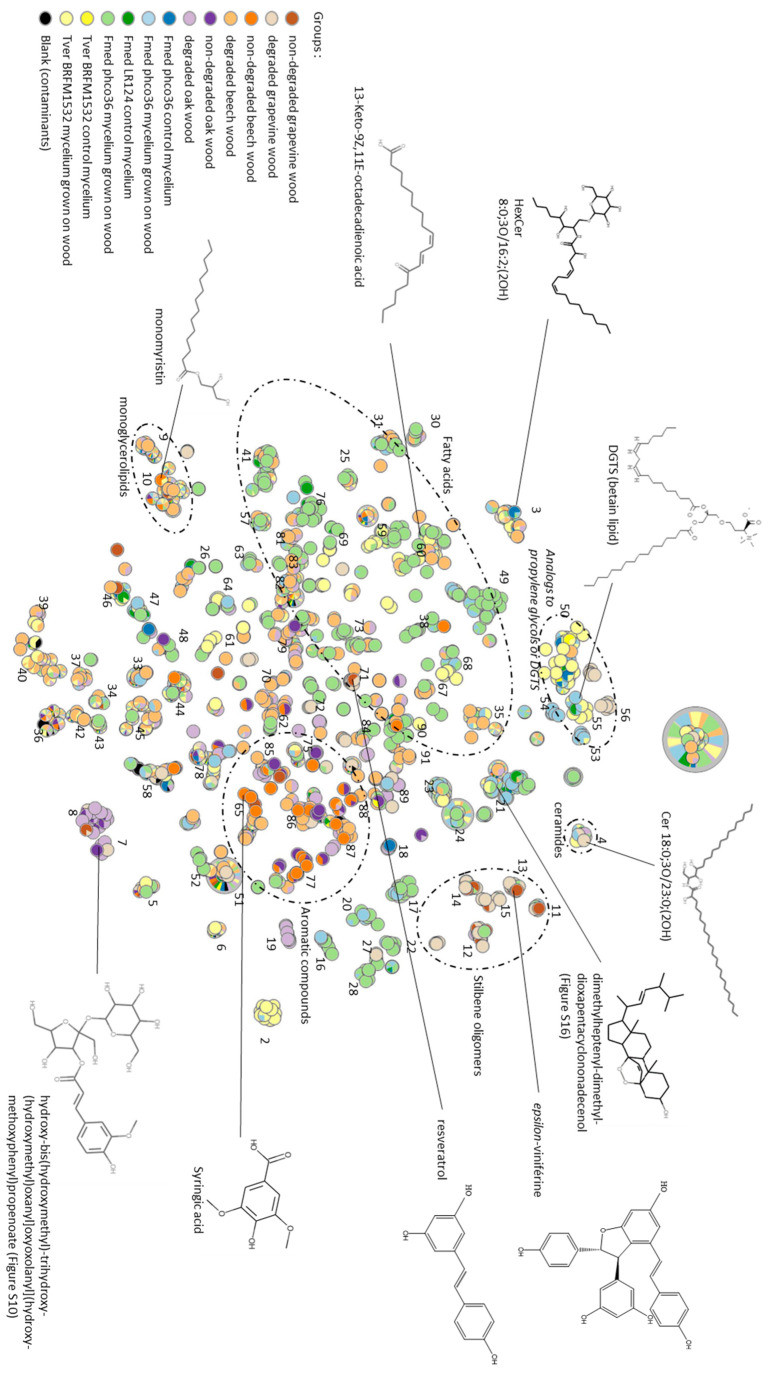
t-SNE graphical representation of the 3011 selected MS/MS features from all samples (wood and mycelium extracts). Minimal cosine score was set at 0.75. The numbers correspond to clusters constructed with MetGem. Circles were constructed manually according to GNPS and MS-DIAL databases annotation results. Nodes colors correspond to the relative proportions of each feature in the sample groups: non-degraded grapevine, beech and oak wood, respectively, correspond to control grapevine, beech and oak wood extracts; degraded grapevine, beech and oak wood correspond to extracts of woods after exposure to all three strains; for each strain, mycelium grown on the wood group corresponds to all mycelia grown on the three wood species. Nodes in black or with black are features also found in the blank sample (methanol alone) and correspond to contaminants.

**Figure 11 jof-09-00536-f011:**
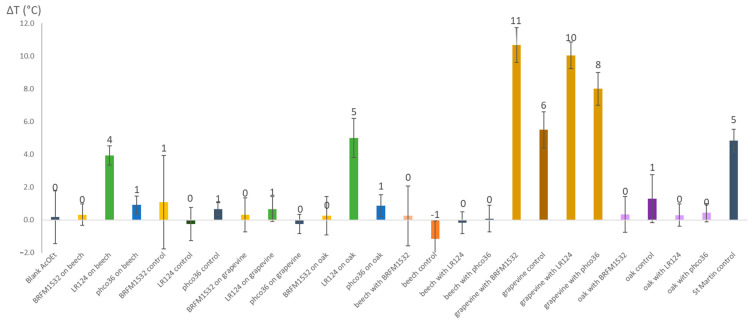
GSTO2S denaturation temperature shift (ΔT) in the presence of blank extract (Blank AcOEt), mycelium extracts (BRFM1532 on beech to phco36 on oak) or wood extracts (from beech with BRFM1532 to St Martin control) relative to control denaturation temperature (in the presence of DMSO only).

## Data Availability

Not applicable.

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
