# Peer review of "Wood Degradation by *Fomitiporia mediterranea* M. Fischer: Exploring Fungal Adaptation Using Metabolomic Networking"

_jof, 2023, doi:10.3390/jof9050536_

Round 1

Reviewer 1 Report

The work is devoted to the study of the decomposition of various types of wood by white rot fungi.The work was done at a high methodological level and is of scientific interest. However, the authors should clarify some points. 

p.4 Lines 143-145 It is not entirely clear how the separation of mycelium, wood and malt agar occurred. Does the mycelium grow inside the wood? If it grow, how was it taken into account in the work? It should have been more detailed.

p.6 Line 240 "...5 μL of GSTO2S purified from Trametes versicolor cultures..." It would be necessary to describe the method for obtaining the enzyme or give a referenсe to the work describing the method.

Figure 5. The error bars cover the dots between the numbers. For example a) Lignin 1.1 looks like 11. It should have been corrected in the figure.

Author Response

Thank you for reviewing our article.

Please find hereunder answers to your comments and questions :

p.4 Lines 143-145 : mycelium grown indeed inside the wood, as well as on the surface. Thus, wood and the part of the mycelium grown inside the wood were analyzed together, and mycelium grown outside the wood blocks were separated. This was taken into account for the interpretation of the results.

p.6 Line 240: the method was indeed already described and the appropriate reference was added

Figure 5: the position of the numbers was corrected

Reviewer 2 Report

Comment_1

The article under review is striking in the abundance of modern methods used to study the development of one of the wood-destroying fungi that parasitizes on grapevine trunk. The results obtained are of high scientific novelty and relate to many aspects of physiology not only of Fomitiporia mediterranea, but also of wood-destroying fungi in general. We can confidently predict that its publication will be of interest to many specialists. However, when reading the manuscript of the article, several questions arise. The main one concerns the method of work. The studies were carried out using dead wood of grapevine, oak and beech, and the following questions arise: 1) is it possible to speak about the interaction of fungus and wood in this case; 2) what extent can the data obtained be used to understand the processes of interaction between living vine, oak, beech wood with Fomitiporia mediterrane and other wood-destroying fungi? Maybe it would be more logical to use samples of wood affected by fungi in a natural way? There are questions about the design of the manuscript. In particular, in the "Results" section, there are repeatedly materials related to the methods used in the work, as well as references to literary sources. For example, in lines 271,326 there are links to publications numbered 16 and 43, respectively. Repetitions concerning the methods of work can be seen, for example, in lines 286-290 and 315-316. There are complaints about figure 9: in the form in which it is presented in the manuscript, I believe it cannot be published, since it is impossible to understand what is shown on it! 

Author Response

Thank you for reviewing our article.

Please find hereunder answers to your comments and questions :

1) Indeed, only the effect of the fungi on the wood, and not an interaction between wood and the fungus can be observed with this methodology (this formulation was corrected in the text).

2) Indeed, it would be very interesting to study wood colonization by the fungus in living wood. The aim of this study was to investigate firstly Fmed wood colonization mechanisms on woods showing different structural and chemical initial compositions. Objectives in the introduction were reformulated. Further studies taking into account wood responses or interactions with other microorganisms would be very interesting, as suggested in the conclusion of this work.

Regarding repeatedly materials or references in the results section: references were discarded and moved either in the materials and methods or in the discussion sections. Short repetitions concerning the method were included in the results as a reminder to help the reader. Those repetitions were shortened.

Figure 9 caption and features annotation in the figure were corrected to be better legible.